# The Association between Differentiation of Self and Life Satisfaction among Chinese Emerging Adults: The Mediating Effect of Hope and Coping Strategies and the Moderating Effect of Child Maltreatment History

**DOI:** 10.3390/ijerph19127106

**Published:** 2022-06-09

**Authors:** Xiamei Guo, Jingwen Huang, Yuexia Yang

**Affiliations:** Institute of Psychology, School of Public Affairs, Xiamen University, Xiamen 361000, China; 33420201150835@stu.xmu.edu.cn (J.H.); 33420211150906@stu.xmu.edu.cn (Y.Y.)

**Keywords:** differentiation of self, hope, coping strategies, life satisfaction, child maltreatment

## Abstract

Background: Differentiation of self (DoS) is a core construct in Bowen family systems theory. At the interpersonal level, it represents the capacity to maintain rational thinking rather than reacting emotionally, especially while under stress. Previous studies have demonstrated the positive association between DoS and life satisfaction. The current study aims to investigate the mediating roles of hope and coping strategies on this association, and whether the mediation mechanism was moderated by participants’ experience of child maltreatment. Methods: The current sample consisted of 447 Chinese college students recruited from three Chinese universities. DoS, life satisfaction, hope and coping strategies, as well as childhood maltreatment history, were measured via self-report. Structural equation modeling was used to test the proposed mediation and moderation effect. Results: Participants who were maltreated in childhood (*n* = 149) exhibited significantly lower levels of DoS, hope, and positive coping strategies than the comparison group (*n* = 298) at baseline and lower life satisfaction at the 3-month follow-up. Structural equation modeling analysis showed that coping strategies mediated the association between DoS and life satisfaction for both the maltreated and comparison groups. Hope appeared to have a significant mediating effect only among those in the comparison group. Conclusions: The current findings lend support to Bowen’s theoretical statement regarding the role of DoS on psychological well-being, with an incorporated viewpoint of Snyder’s hope theory.

## 1. Introduction

Bowen family systems theory (BFST) [1] is one of the most influential theoretical frameworks for guiding researchers toward a systemic understanding of the interplay between stress, emotional regulation, and health. Differentiation of self (DoS) is one of the central constructs in BFST [1,2], which refers to the capacity to regulate one’s emotional reaction under stress and engage in thoughtful and rational examination to cope with problems [3]. Several studies on Western populations have connected DoS to beneficial developmental outcomes such as psychological well-being, physical health, and couple adjustment [4,5]. The purpose of this study is to investigate the relationship between DoS and life satisfaction in a group of Chinese emerging adults. In particular, the current study aims to discover if this association is mediated by hope and coping strategies, as well as to see if the mediating mechanisms differ between those who were abused or neglected during childhood and those who did not.

### 1.1. Differentiation of Self and Life Satisfaction

Differentiation of self, according to BFST, is the internalization of family relationships, which is defined by a balance of autonomy and connectedness [3]. Bowen points out that people who are better differentiated have greater functioning on both the interpersonal and the intrapersonal levels [1]. DoS refers to the ability to strike a balance between maintaining close relationships with important others and developing an autonomous sense of self in close relationships on the interpersonal level [2]. The postulated positive link between DoS and couple adjustment has received a lot of empirical support [6,7]. On the intrapersonal level, DoS refers to the ability to separate rational reasoning from feelings or emotional reactions when under stress [3]. In other words, people with higher DoS are more capable of thinking rationally, coping with uncertainty, and remaining relatively calm while encountering life stressors, whereas people with lower DoS tend to be overwhelmed by their emotions. Several studies have shown that higher levels of DoS are associated with higher levels of life satisfaction [8,9]. Taken together, Bowen’s statement regarding the role of DoS in the trajectory of individual and couple adjustment has received a great deal of empirical support, with some studies confirming the cross-cultural validity with non-US samples [10]. However, evidence regarding the association between DoS and life satisfaction with Chinese samples is still limited.

### 1.2. The Mediating Role of Hope and Coping Strategies

Snyder defines hope as a positive motivational state based on an individual’s assessment of his or her own capacity to develop pathways toward a goal and to direct goal-oriented energy to pursue it [11]. Contrary to the viewpoint that hope is solely an emotion [12], Snyder emphasizes the cognitive processes of hope [11]. In this realm of research, a variety of studies have linked hope to positive psychological adjustment including resilience and life satisfaction among people either with severe diseases or traumatic experiences [13,14] or without such experiences [15,16]. Childhood is the critical stage for the development of the two cognitive parts of hope, pathways, and agency thinking, according to Snyder’s hope theory [11]. Children’s hope-related traits could be shaped through the interaction with primary caretakers. It is also feasible that the effect of hope might link DoS levels to individual adjustment. DoS is a quality that represents one’s capacity to retain rational thinking and suppress emotional emotions, which is created with everyday family contact. Indeed, several researchers discovered that hope moderated the relationship between DoS and good qualities, including dedication to social justice [17]. However, it has seldom been explored whether hope mediates the relationship between DoS and life satisfaction, especially those among non-US samples.

According to Bowen, one’s DoS level is related to one’s ability to cope with life stress [1]. People who are highly differentiated cope better with stresses and have stronger problem-solving abilities [18,19], whereas those who are poorly differentiated may rely on emotion-oriented coping [20]. Snyder also notes that hope and coping strategies are positively associated [21]. Chang reported that individuals who have higher levels of hope have better problem-solving abilities than those who have lower levels of hope [22]. As a result, both hope and coping strategies were expected to mediate the relationship between DoS and life satisfaction in the current study.

### 1.3. The Moderating Effect of Child Maltreatment

Child maltreatment, such as child abuse and neglect, has long been recognized as a significant risk factor for psychological well-being [23,24]. Child maltreatment is viewed as a facet of family functioning associated with insufficient emotional separation between family members, as well as an inability to act on principles rather than feelings, according to the BFST concept [25]. Snyder also considers that being ignored and abused as a child is one of the primary causes of children’s hopelessness [11]. There is some evidence that adverse early life events, such as growing up with alcoholic family members or experiencing ongoing military conflict as a youngster, are associated with lower levels of DoS [26,27]. Furthermore, evidence suggests that violence and trauma may impair the development of coping mechanisms [28]. However, it has rarely been explored if child abuse interferes with the relationship between DoS and life satisfaction by diminishing their hope and/or coping skills.

A great amount of research has investigated the moderation and mediation mechanisms between child maltreatment and child development, with an aim to shed more light on the prevention and intervention effort for this vulnerable subpopulation of children [24]. Nevertheless, the majority of current studies regarding child maltreatment were conducted in developed countries, and more research efforts from developing countries are needed [29]. The most recent estimates of child abuse and neglect exposure among Chinese youth ranged from 20% to 47% [30,31], which was comparable to other nations’ estimates [32]. It is critical to research the lived experiences of Chinese youth who have been maltreated, as well as the long-term effects of maltreatment on their adult well-being.

### 1.4. The Current Study

With an aim to broaden the knowledge of BFST and hope theory, in the current study, we investigated the prospective association between DoS and life satisfaction among a sample of Chinese college students. We hypothesized that (1) DoS would be positively associated with life satisfaction in 3 months; (2) the association between DoS and life satisfaction would be mediated through hope and coping strategies. The current study also hypothesized that (3) participants who were maltreated during childhood would exhibit lower levels of DoS, hope, positive coping strategies, and life satisfaction, as well as higher levels of negative coping strategies; and (4) the mediating effects would be different between participants who were maltreated during childhood and their counterparts who were never maltreated. Given the consistent findings in the literature regarding the relationship between self-esteem and childhood psychological maltreatment, as well as subjective well-being among Chinese emerging adults [32,33], in the current study, we included self-esteem as a control variable.

## 2. Materials and Methods

### 2.1. Data Collection and Research Participants

Data were collected from three universities in a southeastern province in China from March to September 2020. A total of 453 students were assessed twice with a 3-month interval using the Wenjuanxing online survey tool. Six respondents had invalid answers at baseline. Thus, the final sample size was 447. The current sample consisted of 107 male emerging adults and 340 female emerging adults, with the average age being 20.05 years old (S.D. = 1.61). Less than half of the current sample was the only child in the family (*n* = 193), and fewer than one-sixth of the sample was from non-intact families (*n* = 69). Regarding the parental educational level, 42.51% of the sample (*n* = 190) reported that their father had a degree of junior high school or below, 37.81% of the fathers (*n* = 169) had a high school degree, and 19.69% (*n* = 88) had a college or a postgraduate degree. In addition, 50.34% of participants’ mothers (*n* = 225) had a degree in junior high school or below, one-third had a degree in high school (*n* = 149), and 16.33% of participants (*n* = 73) reported that their mother had a bachelor or master degree. The number of participants who completed the follow-up assessment was 432; thus, the follow-up rate was 96.64%.

Participants of the current study were recruited through advertisements. Before the assessment, all participants were fully informed about the objectives, content, procedures, and confidentiality of the current study and were notified that they had the choice to withdraw at any time. Written consent was obtained from all participants. The baseline assessment took about 20–30 min, and the follow-up assessment took about 10 min to finish. Each participant received 30 RMB for each assessment as compensation for their time.

### 2.2. Measurements

DoS was measured with the Chinese version of the Differentiation of Self Inventory (DSI) at the baseline assessment [34,35]. The Chinese version of DSI is composed of five subscales: Emotional reactivity, Taking “I” position, emotional cutoff, fusion with others, and Fusion with Family. The DSI includes 40 items on a 6-point Likert scale ranging from 1 (*not at all true of me*) to 6 (*very true of me*). In the current study, the Cronbach’s α of the subscale scores ranged from 0.66 to 0.86. The five subscales of DSI were used as indicators of a latent variable of DoS in the structural equation modeling analysis.

Hope was assessed using the Hope Scale (HS) at the baseline assessment [36]. It includes eight items measuring the two components of hope—namely, pathway thinking and agency thinking. Each item is rated on a 4-point Likert scale ranging from 1 (*definitely false*) to 4 (*definitely true*). The total score of HS was used in this study, with higher scores indicating higher levels of hope. The Cronbach’s α was 0.92 in the present sample.

Coping strategies were measured using the Simplified Coping Style Questionnaire (SCSQ) at the baseline assessment [37]. The SCSQ is based on the Ways of Coping questionnaire by Folkman and Lazarus [38] and consists of two subscales—problem-oriented/positive coping and emotion-oriented/negative coping. The SCSQ has 20 items on a 4-point Likert scale and is commonly used in China. In the current study, the Cronbach’s α coefficients were 0.79 and 0.70 for positive coping and negative coping subscales, respectively. The two subscales of SCSQ were used as indicators of a latent variable of coping in the structural equation modeling analysis.

The Chinese version of the short form of the Childhood Trauma Questionnaire (CTQ-SF) was used to measure the maltreatment experience before the age of 16 [39,40]. The CTQ-SF has 25 clinical items that are used to measure the five maltreatment constructs, including physical, emotional, and sexual abuse, and physical and emotional neglect. The Cronbach’s α coefficients ranged from 0.72 to 0.87 in the current study. The cutoffs of each subscale are (1) physical abuse ≥10, (2) emotional abuse ≥13, (3) sexual abuse ≥8, (4) physical neglect ≥10, and (5) emotional neglect ≥15. Those for whom all five subscale scores were below the cutoffs were categorized as the comparison group with no child maltreatment experience. The ones with at least one subscale score above the cutoff were categorized as the maltreated group. Based on the cutoff criteria of CTQ-SF, 149 youth in the current sample experienced at least one form of neglect and abuse before the age of 16, whereas 298 youth had no child maltreatment experience.

Self-esteem was assessed by the Rosenberg Self-esteem Scale [41], a widely used measurement of self-esteem, at the 3-month follow-up. This instrument has 10 items rated on a 4-point Likert scale. The sum score was used to represent the level of self-esteem given that this scale generates one factor. The Cronbach’s α coefficient was 0.89 in the current study.

Life satisfaction was assessed with the Satisfaction with Life Scale (SWLS) at the 3-month follow-up [42]. It comprises 5 items on a 7-point Likert scale ranging from 1 (*strongly disagree*) to 7 (*strongly agree*). The sum of all item scores was used in the analysis, with higher scores indicating higher levels of overall life satisfaction. The Cronbach’s alpha of SWLS in this study was 0.86.

### 2.3. Analytic Strategies

Descriptive analysis was conducted with SPSS version 22 (IMB Corp, Armonk, NY, USA). Independent sample *t*-tests were used to compare the differences in their DoS, hope, coping strategies, and life satisfaction based on gender, family condition (intact versus non-intact), and childhood maltreatment experience. Correlation analyses were used to examine the relations among the variables of interest. Structural equation modeling analysis was used to test the hypothesized relationships between DoS and life satisfaction as well as the mediating and moderating effects with AMOS 24.0 and the maximum likelihood method of parameter estimation. The model shown in Figure 1 was analyzed. Demographic characteristics including age, gender, and family structure (intact or non-intact) were included as covariates. Self-esteem was also used as a control variable. The moderating effect of child maltreatment (0 = maltreated group; 1 = comparison group) was tested with the multigroup analysis. An unconstrained model was compared with models with equal measurement weights, equal measurement intercepts, structural weights, means, etc. AMOS provides chi-square test results for the nested models. The root-mean-square error of approximation (RMSEA) and the Comparative Fit Index (CFI) were used to evaluate the goodness of fit of the one model with the best fit among the nested models. RMSEA values less than 0.06 and CFI values greater than 0.95 indicate a good fit between data and models, whereas RMSEA between 0.06 and 0.08 with a CFI between 0.90 and 0.95 indicates an acceptable fit [43]. If a mediator was significantly associated with an independent variable (path *a*) and a dependent variable (path *b*) simultaneously, the significance of the mediation effect (the product of the point estimate of paths *a* and *b*) was estimated with a bootstrapping procedure [44]. Specifically, 2000 bootstrap samples from the original dataset were formed through random sampling with replacement. The mediation effect was considered statistically significant at the 0.05 level if its 95% confidence interval did not include zero.

## 3. Results

### 3.1. Basic Information of the Participants

Means and standard deviations of the variables are presented in Table 1. The skewness and kurtosis values for each variable except the CTQ-SF were between −1 and +1; therefore, the data were considered to have a normal distribution. In the current sample, females had significantly lower scores on emotional reactivity (*t*(444) = 3.20, *p* < 0.01) and “I” position from DSI (*t*(444) = 3.47, *p* < 0.01), as well as lower levels of hope (*t*(444) = 2.21, *p* < 0.05) than males. Age was significantly correlated with emotional cutoff (*r*(444) = 0.12, *p* < 0.05) and fusion with others (*r*(444) = 0.12, *p* < 0.05). There was no significant difference between those from intact families and their counterparts from non-intact families. Independent sample *t*-tests showed that participants in the maltreated group exhibited significantly lower levels of hope (*t*(444) = −4.56, *p* < 0.001), positive coping skills (*t*(444) = −2.93, *p* < 0.01), and self-esteem (*t*(431) = −4.17, *p* < 0.001) than the comparison group at baseline, and significant lower life satisfaction (*t*(287.48) = −5.59, *p* < 0.001) at the 3-month follow-up. In addition, those in the maltreated group also scored significantly lower in four of the five DSI subscales, including emotional reactivity, I position, emotional cutoff, and fusion with others (*ps* < 0.05). Results of correlation analysis are presented in Table 2.

### 3.2. Structural Equation Modeling Analysis

The full model was analyzed using the multigroup analysis function in AMOS 24 with the child maltreatment history as the grouping variable. In this way, the parameter estimates were compared between the maltreated group and the non-maltreated group. Preliminary analysis showed that the path coefficients of demographic variables, including age, gender, and family structure were not significant for either group. Therefore, these paths were removed, and only the covariance paths between demographic variables and independent variables (DoS and self-esteem) were kept in the final model. Multigroup analysis showed that the model with equal measurement weights did not fit the data worse than an unconstrained model (χ^2^(8) = 10.29, *p* > 0.05) but fit the data better than a model with equal measurement intercepts (χ^2^(9) = 31.35, *p* < 0.01). Therefore, the measurement weights model was retained as the final model, which showed an adequate fit to the data (CFI = 0.93, RMSEA = 0.050, 90% CI (0.041,1 0.060)). Path coefficient results of the measurement weights model are presented in Table 3. In the measurement weights model, four regression weights were free to vary between the two subgroups of participants, which represented the associations between DoS and coping, hope and coping, hope to life satisfaction, and self-esteem to life satisfaction. The corresponding paths are presented with dashed lines in Figure 1. The other paths, which were restrained to be equal between the two groups, are presented with solid lines in Figure 1.

Among the paths that were set to be equal across the two groups, DoS and hope were significantly associated (*b* = 1.34, S.E. = 0.11, *z* = 12.20, *p* < 0.001), as were coping and life satisfaction (*b* = −1.64, S.E. = 0.67, *z* = −2.45, *p* < 0.05). The direct effect of DoS on life satisfaction was equally significant for both groups (*b* = 0.77, S.E. = 0.35, *z* = 2.18, *p* < 0.05). Among the paths that were free to vary across the two groups, the associations between DoS and coping strategies were positive for the maltreated group (*b* = 0.30, S.E. = 0.09, *z* = 3.37, *p* < 0.001) and the comparison group (*b* = 0.45, S.E. = 0.09, *z* = 5.01, *p* < 0.001). Hope was positively associated with coping strategies for both groups (*b* = 0.10, S.E. = 0.05, *z* = 2.10, *p* < 0.05; *b* = 0.11, S.E. = 0.04, *z* = 2.57, *p* < 0.05). The relationship between hope and life satisfaction were not significant among the maltreated group (*p* = 0.058) but in fact reached statistical significance among the comparison group (*b* = 0.39, S.E. = 0.10, *z* = 3.71, *p* < 0.01). The overall model explained 56% and 51% of the total variance in life satisfaction for the maltreated and comparison groups, respectively.

Bootstrapping estimation showed that the mediation effect of coping strategies on the association between DoS and life satisfaction was significant for the maltreated group (95% CI: (−1.21, −0.07)) and the comparison group (95% CI: (−1.45, −0.14)). The mediation effect of hope was significant among the comparison group (95% CI: (0.28, 0.81)) but was insignificant among the maltreated group (95% CI: (−0.01, 0.46)). There was also a significant chain mediation through hope and coping among the non-maltreated group (95% CI: (−0.79, −0.02)).

## 4. Discussion

In the current study, we investigated the influence of one’s DoS on life satisfaction and the mediating effects of hope and coping strategies, as well as the moderating effect of child maltreatment history among a sample of Chinese college students. Independent *t*-test results showed that those who were maltreated during childhood did exhibit significantly lower levels of DoS, hope, positive coping strategies, and life satisfaction than their counterparts without maltreated experience, as hypothesized. DoS was found to have both direct and indirect effects on life satisfaction regardless of participants’ child maltreatment experience in the structural equation modeling analysis. The mediating role of coping was significant for both groups, whereas the mediating role of hope was only significant for those without maltreatment experience.

The current study found that DoS was positively and prospectively associated with life satisfaction among a sample of Chinese young adults, which added to the line of research on the cross-cultural validity of the function of DoS. For instance, Tuason and Friedlander reported that DoS was associated with psychological well-being and anxiety in a Philippine sample [45]. In addition, Rodrigues-Gonzalez et al. reported that higher levels of DoS were linked with lower rates of physical illnesses and psychological symptoms among a sample of Spanish adults [10]. Among the core constructs of BFST, the proposed function of DoS has received consistent empirical support in general [4,35]. This study also expands the literature on BFST by examining the influence of child maltreatment history on DoS. Bowen considers child maltreatment as a facet of negative family functioning, which would influence the developing process of children’s DoS [1]. This notion was supported by the current findings that those who were maltreated in childhood exhibited significantly lower levels of DoS as well as lower levels of life satisfaction than their counterparts. The comparison of the mediation mechanism between the two subsamples based on their maltreatment history provided further information regarding how child maltreatment might impact the association between DoS and life satisfaction.

Regardless of participants’ maltreatment history, coping strategies were found to be a significant mediator in the association between DoS and life satisfaction in this study. From the perspective of BFST, individuals who are well-differentiated are more likely to employ problem-focusing coping strategies rather than avoidant or emotion-oriented coping strategies [18]. This study is in line with the BFST viewpoint that DoS and coping strategies are positively associated. However, the current finding of a negative association between coping strategies and life satisfaction contradicts the results of previous studies, in which a positive association was found [46]. Muyan-Yilik and Demir reported a similar pattern to the current findings with a sample of Turkish college students [47], according to which avoidance-oriented coping, rather than problem-oriented coping, was positively associated with life satisfaction. The current findings might also reflect unique challenges faced by college students. Conley et al. found that college students experienced decreased self-esteem and increased avoidant coping during the first two years in college [48]. Therefore, their negative coping strategies might provide them with a temporary escape from life stressors, which, in turn, was associated with higher levels of life satisfaction in the short term. Future research is needed to further investigate the association between coping strategies and long-term well-being among this subpopulation to provide an in-depth explanation.

Among the subsample of Chinese emerging adults without child maltreatment experience, hope was found to be a significant mediator in the association between DoS and life satisfaction. In addition, a significant chain mediation through hope and coping was found among the non-maltreated subsample. In other words, hope served as a protective factor of life satisfaction among this subgroup of emerging adults, which was consistent with previous research [47]. The current findings also suggested that the history of child maltreatment might weaken the relation between hope and life satisfaction among Chinese emerging adults, and the protective mediation effect of DoS on life satisfaction through hope diminished. This finding might shed some insight on the clinical practice with clients who were maltreated in childhood. From the BFST perspective, family therapists would focus on assisting their clients with abuse or trauma experiences to strengthen their capacities for healthy interpersonal connection through promoting their differentiation [25]. The current findings suggested that it would also be important to rebuild their cognitive ability of hope and repair its protective mechanism when working with clients with child maltreatment history.

Several limitations should be noted when interrupting the results. First, the current study only surveyed a sample of college students from a southeastern province in China. Over three-fourths of the current sample were females, possibly due to the fact that male Chinese college students were more reluctant to participate in surveys. Therefore, the generalizability of the current findings is limited. Second, although the current data were not merely cross-sectional, there was only a 3-month span between the two assessments. Future research with longitudinal data is needed to further examine the long-term influence of DoS on well-being. Third, the current study consolidated five forms of child abuse and neglect into one maltreated group due to the small sample size. Future research needs to further compare the effect of different types of abuse and neglect with a large sample. Last but not least, the current sample size was relatively small considering the group with child maltreatment history, which also restricted the ability to account for more variables that might influence one’s DoS and life satisfaction. Other influential variables that were not accounted for in the current study, such as personality traits and concurrent stress levels, may need further investigation.

## 5. Conclusions

Bowen proposes that the ability to maintain rational thinking when facing life stressors plays an important role in individual adjustment. Snyder emphasized the positive motivational state and cognitive processes of hope on individual development. The current study showed that BFST’s statement regarding the function of DoS could be generalized to Chinese emerging adults. Moreover, the current study examined the mediating role of hope and coping on the association between DoS and life satisfaction. In particular, the current findings suggested that child abuse and neglect experience might impede the positive association between DoS and life satisfaction by weakening the mediation effect of hope. The mediation effect of coping strategies, on the other hand, did not differ in the history of child maltreatment. Future research on BFST is still needed, especially those focusing on at-risk subpopulations. Clinicians and practitioners might need to target improving the agency and pathway thinking of hope among those who were maltreated to promote their life satisfaction.

## Figures and Tables

**Figure 1 ijerph-19-07106-f001:**
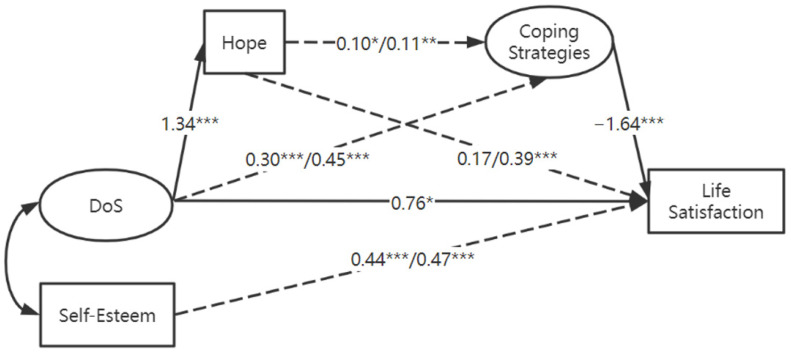
Path coefficients from the final model in the structural equation modeling with multigroup analysis. The dashed line represented the paths in which the path coefficients were free to vary between the two groups, and the solid lines represented the paths in which the coefficients were constrained to be equal between the two groups. The coefficients on the left of the forward slash are from the maltreated group, and those on the right of the forward slash are from the non-maltreated group. * *p* < 0.05. ** *p* < 0.01. *** *p* < 0.001.

**Table 1 ijerph-19-07106-t001:** Demographic information and descriptive analysis.

	*n* (%)	Mean (Standard Deviation)
Gender		
Male	107 (23.94%)	
Female	340 (76.06%)	
Age		20.05 (1.61)
Parental marriage status		
Currently married	378 (84.56%)	
Others (divorced, deceased, or never married)	69 (15.44%)	
Self-esteem		27.30 (4.80)
Positive coping strategies		23.35 (5.42)
Negative coping strategies		10.07 (4.11)
Hope		45.50 (9.44)
Life satisfaction		20.45 (5.58)
Differentiation of Self Inventory subscales		
Emotional reactivity		37.05 (9.91)
“I” position		38.95 (7.07)
Emotional cutoff		29.56 (5.89)
Fusion with others		17.59 (4.58)
Fusion with family		11.30 (3.27)
Childhood Trauma Questionnaire subscales		
Physical abuse		5.66 (1.63)
Emotional abuse		7.06 (2.71)
Sexual abuse		5.50 (1.35)
Physical neglect		9.21 (1.61)
Emotional neglect		9.64 (4.40)

**Table 2 ijerph-19-07106-t002:** Bivariate correlations among the study variables.

Variables	1	2	3	4	5	6	7	8	9	10
1. Self-esteem	1									
2. DSI-emotional reactivity	0.44 **	1								
3. DSI-“I” position	0.50 **	0.45 **	1							
4. DSI-emotional cutoff	0.39 **	0.48 **	0.14 **	1						
5. DSI-fusion with others	0.43 **	0.72 **	0.54 **	0.42 **	1					
6. DSI-fusion with Family	0.02	0.30 **	0.05	−0.01	0.18 **	1				
7. Hope	0.33 **	0.04	0.31 **	0.05	0.03	−0.19 **	1			
8. Positive coping	0.30 **	0.12 *	0.31 **	0.22 **	0.14 **	−0.20 **	0.33 **	1		
9. Negative coping	−0.30 **	−0.39 **	−0.32 **	−0.26 **	−0.37 **	−0.10 *	−0.10 *	0.05	1	
10. Life satisfaction	0.62 **	0.27 **	0.39 **	0.26 **	0.26 **	−0.14 **	0.41 **	0.30 **	−0.13 **	1

DSI: Differentiation of Self Inventory. * *p* < 0.05. ** *p* < 0.01.

**Table 3 ijerph-19-07106-t003:** Parameter estimates from structural equation modeling analysis.

Path			Maltreated Sample (*n* = 149)				Non-Maltreated Sample (*n* = 298)			
			Estimate	S.E.	C.R.	*p*-Value	Estimate	S.E.	C.R.	*p*-Value
DoS	→	DSI-IP	1				1			
DoS	→	DSI-EC	0.487	0.069	7.018	***	0.487	0.069	7.018	***
DoS	→	DSI-ER	0.94	0.098	9.541	***	0.94	0.098	9.541	***
DoS	→	DSI-FO	0.458	0.043	10.676	***	0.458	0.043	10.676	***
DoS	→	DSI-FF	−0.069	0.036	−1.899	0.058	−0.069	0.036	−1.899	0.058
Coping	→	SCSQ-positive coping	1				1			
Coping	→	SCSQ-negative coping	−0.536	0.093	−5.784	***	−0.536	0.093	−5.784	***
DoS	→	Hope	1.343	0.11	12.203	***	1.343	0.11	12.203	***
DoS	→	Coping	0.298	0.089	3.336	***	0.447	0.089	5.01	***
Hope	→	Coping	0.097	0.046	2.095	0.036	0.105	0.041	2.574	0.01
Coping	→	LS	−1.644	0.673	−2.445	0.014	−1.644	0.673	−2.445	0.014
Hope	→	LS	0.168	0.088	1.899	0.058	0.385	0.104	3.709	***
Self-esteem	→	LS	0.443	0.119	3.717	***	0.465	0.082	5.638	***
DoS	→	LS	0.756	0.345	2.193	0.028	0.756	0.345	2.193	0.028

DoS: Differentiation of Self; DSI: Differentiation of Self Inventory; IP: “I” position; EC: emotional cutoff; ER: emotional reactivity; FO: fusion with others; FF: fusion with family; SCSQ: the Simplified Coping Style Questionnaire; LS: life satisfaction. *** *p* < 0.001.

## Data Availability

The datasets used and analyzed in this study are available from the corresponding author on reasonable request.

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
