# Peer review of "The Association between Differentiation of Self and Life Satisfaction among Chinese Emerging Adults: The Mediating Effect of Hope and Coping Strategies and the Moderating Effect of Child Maltreatment History"

_ijerph, 2022, doi:10.3390/ijerph19127106_

Round 1
Reviewer 1 Report
Thank you very much for the opportunity to review the manuscript entitled " The Association Between Differentiation of Self and Life Satisfaction among Chinese Emerging Adults: The Mediating effect of Hope and Coping Strategies and the Moderating Effect of Child Maltreatment History", which was sent to IJERPH.
This research fits into the main topic of the journal.
Despite the fact that this topic is interesting, and I really think that the authors have devoted a lot of effort to their research, this current manuscript still needs improvement.
Firstly, in view of the sensitivity of the hypothesis testing method used to the normal distribution of variables, it is necessary to provide data on checking the variables for the normality of the distribution.
Secondly, there is a lack of clarity in the presentation of the results of Table 3. From the above material, it is not clear what the tested model is as a result. You should also disclose all the designations used in the tables.
Thirdly, it is not clear on what basis, the authors conclude “Results showed that those who were maltreated during childhood did exhibit significantly lower levels of DOS, hope, positive copying strategies, and life satisfaction than their counterparts without maltreated experience as hypothesized” if no comparative study was conducted. At least, there is no reasoned mention of this in the article (for example, using some kind of parametric criterion).
Fourth, it is necessary to clarify on the basis of what the authors use the general coping indicator, what does it mean in Table 3? What is this indicator based on? The same applies to DoS and DSI designations. Perhaps if there was a real structural model in the article, these issues would be partially removed. Since the authors appeal to certain DoS indicators, it is necessary to indicate in the methods the presence of an integral indicator (if any), as well as the Coping integral indicator.
Finally, since the article is devoted to the study of the role of hope and coping in relation to the differentiation of self and life satisfaction, it would be desirable to more clearly spell out this role in the discussion, make a meaningful conclusion and graphically present models (SEM) for both parts of the sample.
This manuscript holds actual value to the readers on IJERPH.
I will be glad to review the revised manuscript.
Reviewer 2 Report
First of all, congratulations to the authors for their article. Just make two comments to the authors:
- The description of the sample speaks of adolescents and the title of emerging adults. I think that the term adolescents should be changed to that of emerging adults, they are two different evolutionary periods, giving very specific psychological characteristics in the case of emerging adulthood that differentiate it from the period of adolescence as indicated by different studies.
- On the other hand, when talking about the limitations of the study in relation to the sample, it must be taken into account that it is a very feminized sample since more than 70% are women.
Round 2
Reviewer 1 Report
The authors have finalized the text of the article. The new version has more clarity and validity.